# STABLE-LoRA: STABILIZING FEATURE LEARNING OF LOW-RANK ADAPTATION

**Yize Wu**[1,2]**, Ke Gao**[1]**, Ling Li**[1,2]**, Yanjun Wu**[1*]
[1]Intelligent Software Research Center, Institute of Software, CAS, Beijing, China
[2]University of Chinese Academy of Sciences, Beijing, China
{wuyize2021,gaoke,liling,yanjun}@iscas.ac.cn

## ABSTRACT

Low-Rank Adaptation (LoRA) is a widely adopted parameter-efficient method for fine-tuning Large Langauge Models. It updates the weight matrix as $W = W_0 + sBA$, where $W_0$ is the original frozen weight, $s$ is a scaling factor and $A$,$B$ are trainable low-rank matrices. Despite its robust empirical effectiveness, the theoretical foundations of LoRA remain insufficiently understood, particularly with respect to feature learning stability. In this paper, we first establish that, LoRA can, in principle, naturally achieve and sustain stable feature learning (i.e., be self-stabilized) under appropriate hyper-parameters and initializations of $A$ and $B$. However, we also uncover a fundamental limitation that the necessary non-zero initialization of $A$ compromises self-stability, leading to suboptimal performances. To address this challenge, we propose Stable-LoRA, a weight-shrinkage optimization strategy that dynamically enhances stability of LoRA feature learning. By progressively shrinking $A$ during the earliest training steps, Stable-LoRA is both theoretically and empirically validated to effectively eliminate instability of LoRA feature learning while preserving the benefits of the non-zero start. Experiments show that Stable-LoRA consistently outperforms other baselines across diverse models and tasks, with no additional memory usage and only negligible computation overheads. The code is available at https://github.com/Yize-Wu/Stable-LoRA.

## 1 INTRODUCTION

Low-Rank Adaptation (LoRA) (Hu et al., 2022) is an effective and widely adopted parameter-efficient method for fine-tuning Large Language Models (LLMs). Unlike full fine-tuning, which updates all model parameters, LoRA freezes the original weight matrix $W_0$ and introduces two low-rank trainable matrices, $A$ and $B$, with the weight matrix updated by the multiplication of $A$ and $B$. Formally,

$$W = W_0 + sBA, W_0 \in \mathbb{R}^{m \times n}, A \in \mathbb{R}^{r \times n}, B \in \mathbb{R}^{m \times r}$$

where $s$ is a scaling factor. By choosing $r << \min(m,n)$, the number of trainable parameters is reduced from $mn$ to $(m+n)r$, substantially lowering computation and memory overhead while retaining strong learning capacity.

The effectiveness of LoRA has been demonstrated through massive experiments across various models and tasks, and recent studies (Hayou et al., 2024b; Zhang & Pilanci, 2024; Kalajdzievski, 2023) have also begun to theoretically explore the fine-tuning dynamics of LoRA. However, no prior work has established a theoretical explanation for such robust effectiveness. In this paper, we first provide a theoretical analysis showing that, with appropriate hyper-parameters and initializations of $A$ and $B$, LoRA can naturally achieves stable feature learning with respect to model width $n$ (informally, the learned features scale as $\Theta(n^0)$). Furthermore, once this stability is achieved, it will be sustained throughout the entire training process. Such self-stabilizing property provides a theoretical foundation for the observed effectiveness and robustness of LoRA.

According to the analysis, the ideal initialization for ensuring self-stabilization is to set both $A$ and $B$ to zero. However, this leads to practical issues of saddle-point halting (Zhang & Pilanci,

---

*Corresponding author.

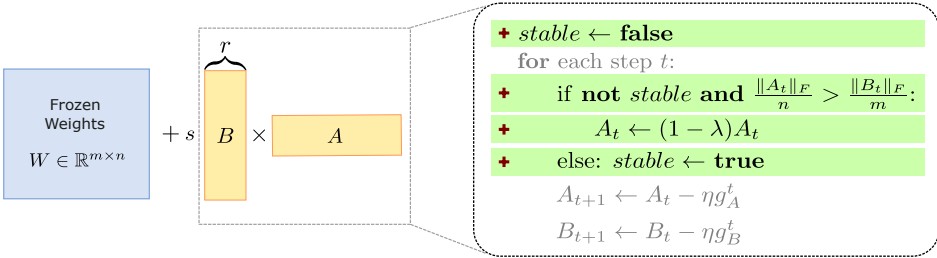

Figure 1: Illustration of Stable-LoRA. The weight-shrinkage operation is emphasized as a patch to the gradient-descent procedure.

2024), information loss and gradient vanishing/explosion (He et al., 2015). The mostly-adopted and theoretically proven-effective (Hayou et al., 2024a) solution is to initialize only $B$ to zero and $A$ non-zero. Nevertheless, we both theoretically and empirically demonstrate that such a non-zero initialization $A_0$ compromises stable feature learning and hence causes suboptimal performances, which motivates the design of novel LoRA optimization strategies.

To address this problem, we propose Stable-LoRA, a weight-shrinkage strategy for LoRA optimization that dynamically enhances the stability of feature learning. We conclude from theoretical perspectives that the initialization-induced instability is a long-term problem whereas others are short-termed. Therefore, Stable-LoRA adopts the non-zero $A_0$ for its benefits and progressively shrinks $A$ as training proceeds. Specifically, a shrinkage ratio $\lambda$ ($0 < \lambda < 1$) is applied to $A$ in the earliest steps of training, updating $A$ according to

$$A_{t+1} = (1 - \lambda)A_t - \eta g_A^t$$

(as shown in Figure 1).This exponential decay diminishes the instability introduced by $A_0$ while still preserving its advantages for early training. Shrinkage stops once the stability condition is satisfied—specifically, when the average norm of $A$ becomes no larger than that of $B$ (see Section 4). We theoretically proved that sufficient shrinkage of $A$ guarantees the prevention of potential instability, thereby ensuring stable feature learning throughout training.

We evaluated Stable-LoRA across different model architectures and tasks, where it uniformly outperforms AdamW and other baselines. Importantly, Stable-LoRA incurs no additional memory usage and introduces only negligible computational overhead—properties that are particularly important in the resource-constrained scenarios where LoRA is most commonly applied.

## 2 PRELIMINARY

### 2.1 FEATURE LEARNING OF LORA

Consider training a weight matrix $W$ with input $Z$, such that the output is $Y = WZ$. In LoRA, the original weight is frozen as $W_0$ and two low-rank trainable matrices $A$ and $B$ are introduced, so that the updated weight becomes $W = W_0 + sBA$, where $s$ is a scaling factor. Given a learning rate $\eta$, the parameter updates at training step $t$ are:

$$A_{t+1} = A_t - \eta g_A^t, B_{t+1} = B_t - \eta g_B^t$$

, where $g_A$ and $g_B$ are the optimizer-processed gradients. The change in output after these updates is given by:

$$
\begin{aligned}
\Delta Y_t &= s(A_t - \eta g_A^t)(B_t - \eta g_B^t)Z - sA_tB_tZ \\
&= -s\eta g_B^t A_t Z - s\eta B_t g_A^t Z + s\eta^2 g_B^t g_A^t Z
\end{aligned}
\tag{1}
$$

. We are particularly interested in the properties of $\Delta Y_t$, as it serves as the "learned feature" at step $t$. Specifically, $\Delta Y_t$ directly influences the inputs to downstream layers and ultimately the model's output, representing the contribution of LoRA updates to the model.

## 2.2 STABLE FEATURE LEARNING

As neural networks continue to scale, understanding their training dynamics with respect to parameter growth becomes increasingly important (Hayou et al., 2024b; Zhang & Pilanci, 2024; Hayou et al., 2019). Much of this analysis has focused on the regime of model width, since in most architectures the parameter count is dominated by width, while depth (i.e., the number of layers) plays a comparatively smaller role. In this regime, a desirable property is that learned features remain "stable" as width increases—they neither explode nor vanish numerically. Such stability is crucial for ensuring meaningful representations can be learned, thereby allowing the model to achieve its full performance potential upon trained tasks.

In the context of LoRA, stable feature learning requires that the output update $\Delta Y_t$ does not scale positively or negatively with model width $n$, otherwise it would explode or vanish as $n$ increases. Formally, this requirement can be expressed as $\Delta Y_t = \Theta(n^0) = \Theta(1)$.

**Definition 1.** *(LoRA stable feature learning) LoRA feature learning is stable, if $\Delta Y_t = \Theta(1)$ for all training steps $t$.*

Note that Definition 1 is slightly different from similar concepts in prior works (Hayou et al., 2024b; Zhang & Pilanci, 2024). It does not require intermediate representations (e.g. $A_t Z$) to individually scale as $\Theta(1)$, but only constrains the final output update. This relaxation is motivated by practical considerations: for an actual (finite) $n$, the scale of intermediate representations can compensate for each other through multiplicative interactions and yield an overall stable output. For example, it is acceptable for components of $\Delta Y_t$ to scale as $U = \Theta(n)$ and $V = \Theta(n^{-1})$, as long as $\Delta Y_t = UV = \Theta(1)$ is ensured.

## 2.3 $\gamma$-FUNCTION

For convenience of notation, we introduce $\gamma$-function to characterize the scaling behavior of variables with respect to the model width $n$. It is defined as follows: for a real-valued scalar variable $v$, we have $v = \Theta(n^{\gamma[v]})$. For a $k$-dimensional tensor variable $\vec{v} = (v_0, \cdots, v_{k-1})$, we define $\gamma[\vec{v}] := \max(v_i, 0 \leq i < k)$, which captures the dominant scaling behavior among its components.

By definition, $\gamma$-function obeys the following properties under element-wise operations:

**Multiplication:** For two real-valued variables $v$ and $v'$, $\gamma[v \times v'] = \gamma[v] + \gamma[v']$

**Addition:** For two real-valued variables $v$ and $v'$, $\gamma[v + v'] = \max(\gamma[v], \gamma[v'])$

With this notation, the condition for stable feature learning can be succinctly expressed as $\gamma[\Delta Y] = 0$.

## 2.4 OPTIMIZED GRADIENT

Modern optimizers such as Adam and AdamW (Kingma & Ba, 2014) are generally preferred over Stochastic Gradient Descent (SGD) in fine-tuning scenarios. These optimizers typically normalize gradients through momentum mechanisms (e.g., exponential moving averages in Adam), which effectively prevents the entries of gradients from becoming excessively small or large. In the following analysis, we assume that the normalized gradients have all entries to be $\Theta(1)$, a condition theoretically justified by the internal dynamics of such optimizers and commonly observed in practice (Hayou et al., 2024b). In the context of LoRA, this assumption applies individually to optimized gradients of each low-rank matrices, i.e., $g_A, g_B = \Theta(1)$.

## 3 LORA IS SELF-STABILIZED

In this section, we present a theoretical analysis of LoRA fine-tuning dynamics, showing that LoRA is self-stabilized with potential appropriate choices of hyper-parameters and initializations $A_0$ and $B_0$.

Recall from Definition 1 and Equation (1) that stable feature learning requires all 3 components of $\Delta Y$ to be $\Theta(1)$. Formally, using the multiplication property of $\gamma$-function and the results that

$\gamma[g_A] = \gamma[g_B] = 0$ (as established in Section 2.4), we have the following contraints:

$$\begin{cases} \gamma[s] + \gamma[\eta] + \gamma[A_t Z] = 0 & (\delta_1 = \Theta(1)) \\ \gamma[s] + \gamma[\eta] + \gamma[B_t] + \gamma[g_A^t Z] = 0 & (\delta_2 = \Theta(1)) \\ \gamma[s] + 2\gamma[\eta] + \gamma[g_A^t Z] = 0 & (\delta_3 = \Theta(1)) \end{cases} \tag{2}$$

Readers may notice that it is sufficient for $\Delta Y_t = \Theta(1)$ if just one component is $\Theta(1)$ and the others are $o(1)$. As $\gamma[\delta_1] \geq \gamma[\delta_3]$ and $\gamma[\delta_2] \geq \gamma[\delta_3]$ always hold (later explained in Section 3.1), it suffices to only justify why $\delta_1$ and $\delta_2$ are restricted to be $\Theta(1)$:

Assume that $\delta_1 = o(1)$. To maintain $\Delta Y_t = \Theta(1)$, we must have $\delta_2 = \Theta(1)$, implying that the output update is dominated by $\delta_2$. This situation corresponds to fixing the matrix $B$ and only training $A$ ($g_B = 0$ in Equation (1)), which is clearly suboptimal compared to training with both matrices. The same argument applies if $\delta_2 = o(1)$. Therefore, for effective and balanced learning, both $\delta_1$ and $\delta_2$ must scale as $\Theta(1)$.

Now we focus on the value of $\gamma[A_t Z]$, $\gamma[B_t]$ and $\gamma[g_A^t Z]$, which ultimately correlate to the choices of $s$ and $\eta$.

## 3.1 VALUE OF $\gamma[g_A^t Z]$, $\gamma[A_t Z]$ AND $\gamma[B_t]$.

We begin by stating an assumption on the value of $\gamma[g_A^t Z]$ and some clarification of its soundness.

**Assumption 1.** *With optimized gradient $g_A^t \in \mathbb{R}^{r \times n}$ and input $Z \in \mathbb{R}^{n \times *}$, we have $\gamma[g_A^t Z] = 1$.*

Consider an extremely simplified optimizer, which normalizes each entry of the gradient to its sign, i.e.,

$$g_A^t = \text{sign}(\frac{\partial L_t}{\partial A}),$$

where $\frac{\partial L_t}{\partial A}$ denotes the raw (non-optimized) gradient of $A$ at step $t$. By the chain rule, we have

$$\frac{\partial L_t}{\partial A} = sB_t^T dY_t \times Z,$$

where $dY_t$ is the gradient of the output $Y_t$. Define

$$S^t = sB_t^T dY_t,$$

so that the gradient becomes

$$\frac{\partial L_t}{\partial A} = S^t \times Z = (S_i^t Z_j)_{ij}$$

Therefore we have

$$g_A^t = \text{sign}(\frac{\partial L_t}{\partial A}) = \text{sign}(S^t \times Z) = \text{sign}(S^t) \times \text{sign}(Z)$$

Hence,

$$g_A^t Z = (\text{sign}(S^t) \times \text{sign}(Z))Z = (\text{sign}(Z)^T Z)\text{sign}(S^t)$$

Since $\text{sign}(Z)^T Z = \Theta(n)$ always holds and $S^t = \Theta(1)$ if it is a stable gradient, we conclude that $g_A^t Z = \Theta(n)$, the same as $\gamma[g_A^t Z] = 1$.

As more sophisticated optimizers generally preserve the sign of gradient (Yang et al., 2013), this assumption is well justified and serves as a foundation for our subsequent analysis.

Equation (2) is then refined as:

$$\begin{cases} \gamma[s] + \gamma[\eta] + \gamma[A_t Z] = 0 \\ \gamma[s] + \gamma[\eta] + \gamma[B_t] + 1 = 0 \\ \gamma[s] + 2\gamma[\eta] + 1 = 0 \end{cases} \tag{3}$$

Next, we analysis the value of $\gamma[A_t Z]$ and $\gamma[B_t]$ by induction. Recall from Section 2.1 and the addition property of $\gamma$-function, we have

$$\begin{cases} \gamma[A_t Z] = \max(\gamma[A_{t-1} Z], \gamma[\eta] + 1) \Rightarrow \gamma[A_t Z] \geq \gamma[\eta] + 1 \\ \gamma[B_t] = \max(\gamma[B_{t-1}], \gamma[\eta]) \Rightarrow \gamma[B_t] \geq \gamma[\eta] \end{cases} \tag{4}$$

, which immediately implies that $\gamma[\delta_1] \geq \gamma[\delta_3]$ and $\gamma[\delta_2] \geq \gamma[\delta_3]$ (used as a conclusion above). It is quite intuitive that $\delta_3$ is less significant than $\delta_1$ and $\delta_2$, as $\delta_3$ is quadratic in the typically small learning rate $\eta$; indeed, this term is often even neglected in some prior analysis (e.g., Yen et al. (2025)).

## 3.2 IMPACT OF $A_0$ AND $B_0$.

Based on the induction relations of Equation (4), we have the following two equivalent condition pairs:

$$\begin{cases} \gamma[A_t Z] = \gamma[\eta] + 1 \iff \gamma[A_0 Z] \leq \gamma[\eta] + 1 \\ \gamma[B_t] = \gamma[\eta] \iff \gamma[B_0] \leq \gamma[\eta] \end{cases} \tag{5}$$

, which indicates that the $\gamma$-values of the components are closely related to the initializations $A_0$ and $B_0$.

Importantly, to satisfy Equation (3) , Equation (5) must both hold or neither: if only one of them is an equation, then we will definitely have $\gamma[A_t Z] \neq \gamma[B_t] + 1$, which leads to an undesirable situation that $\gamma[\delta_1] \neq \gamma[\delta_2]$. Hence, there are only two acceptable cases for $A_0$ and $B_0$:

- **Case 1.** $\gamma[A_0 Z] \leq \gamma[\eta] + 1$ and $\gamma[B_0] \leq \gamma[\eta]$
- **Case 2.** $\gamma[A_0 Z] > \gamma[\eta] + 1$ and $\gamma[B_0] > \gamma[\eta]$

Among them, Case 2 is undesirable because the initial values dominate the training results, overriding contributions from learned updates. In contrast, Case 1 ensures that gradient-based updates govern the learning process. More importantly, if Case 1 is satisfied, the left-hand side conditions of Equation (5) are also satisfied, which in turn ensures that all constraints in Equation (3) are met simultaneously. This leads to a unified expression of $\delta$s:

$$\gamma[\delta_1] = \gamma[\delta_2] = \gamma[\delta_3] = \gamma[s] + 2\gamma[\eta] + 1 \tag{6}$$

Therefore, with appropriate initializations satisfying Case 1, tuning $s$ and $\eta$ such that $\gamma[s] + 2\gamma[\eta] + 1 = 0$ results in $\gamma[\Delta Y_t] = 0$, and stable feature learning will be naturally (without any extra operations) achieved and sustained throughout training, validating its empirical robust effectiveness.

**Theorem 3.1.** *(Self-stability of LoRA) LoRA can naturally achieve and sustain stable feature learning, if the hyper-parameters $s$ and $\eta$ are tuned such that $\gamma[s] + 2\gamma[\eta] + 1 = 0$, and the initializations $A_0$ and $B_0$ satisfy $\gamma[A_0 Z] \leq \gamma[\eta] + 1$ and $\gamma[B_0] \leq \gamma[\eta]$.*

## 4 STABLE-LORA

Case 1 suggests that an ideal initialization strategy is to set both $A_0$ and $B_0$ to zero, ensuring that $\gamma[A_0] = \gamma[B_0] = -\infty$, which guarantees the satisfaction of Case 1 with arbitrary $\eta$. However, while stable feature learning is a necessary condition for effective training, it is not sufficient on its own. With setting $B_0 = 0$ feasible, initializing $A_0 = 0$ introduces two empirical issues: (1) the combination $A = 0$ and $B = 0$ is a saddle point with zero gradient, leading to halting of training; (2) the initial input to $B$ is $A_0 Z = 0$, resulting in complete information loss for learning $B$ and possible gradient vanishing/explosion. The common solution is to set $B_0 = 0$ and sample the entries of $A_0$ from a distribution with $\sigma^2 = n^{-1}$, which addresses both issues and has been theoretically (Hayou et al., 2024a) and empirically (He et al., 2015) shown to be beneficial.

From the perspective of feature learning, this initialization yields $\gamma[B_0] = -\infty < \gamma[\eta]$ for arbitrary $\eta$. According to Theorem 3.1, we must also have $\gamma[A_0 Z] \leq \gamma[\eta] + 1$ to ensure stability. However, due to the non-zero entries in $A_0$, this condition imposes a lower bound on $\eta$: it cannot be arbitrarily small, but must be sufficiently large to absorb the magnitude of $A_0 Z$. In practical scenario where

---

**Algorithm 1** Stable-LoRA

---

**Input:** Learning rate $\eta$, shrink rate $\lambda$, weight decay rate $w$, initializations $A_0$, $B_0$
\# Shrink $A$ if the stable condition is not satisfied
$stable \leftarrow$ **false**
**for** training step $t$ **do**
    **if not** $stable$ **and** $\frac{\|A\|_F}{n} > \frac{\|B\|_F}{m}$ **then**
        $A_t = (1 - \lambda)A_t$
    **else**
        $stable \leftarrow$ **true**
    **end if**
    \# Update parameters with optimized gradients and weight decay
    $A_{t+1} = A_t - \eta g_A^t - \eta w A_t$, $B_{t+1} = B_t - \eta g_B^t - \eta w B_t$
**end for**

---

learning rate is usually small, this condition on $\eta$ is typically violated (as supported by empirical results in Section 5.1). Moreover, this issue cannot be resolved solely by altering initialization or tuning hyper-parameters: adjusting $A_0$ cannot reduce $A_0 Z$ uniformly across the batch, since $Z$ varies while $A_0$ is fixed; tuning $s$ is also ineffective since $s$ is not involved in the condition. Consequently, these limitations motivate the development of novel optimization strategies.

We discover that the problem of instable feature learning is fundamentally different from other issues: it is a long-term problem whereas others are short-termed. While the saddle-point and gradient-vanishing/explosion issues arise only at the beginning of training, they naturally resolve as training proceeds with parameters moving away from the saddle point and meaningful signals being propagated to $B$. In contrast, feature-learning instability occurs at the start if $\gamma[A_0 Z] > \gamma[\eta] + 1$ and persists throughout training due to the induction in Equation (4). This leads to a key idea: instead of modifying $A_0$ from the beginning (which would exacerbate the other two problems), we can gradually reduce its negative impact as the training proceeds. The solution would be optimal if $A_0$ can serve its early-stage positive role while its adverse effects diminish to a desirable level over time.

Based on this, we propose Stable-LoRA, a weight-shrinkage optimization strategy applied to matrix $A$ in earliest steps of training, to mitigate instability of LoRA feature learning. An overview of Stable-LoRA is shown in Figure 1, and the detailed procedure is provided in Algorithm 1. Specifically, at an early step $t$, before parameter updates, $A$ first shrinks as

$$A_{t+1} = (1 - \lambda)A_t - \eta g_A^t$$

where $\lambda$ ($0 < \lambda < 1$) is the shrinkage ratio, after which $A$ continues to be updated with $g_A^t$. Shrinkage is applied at every step until a stable condition is met: $A$ achieves a comparable average norm scale to $B$, i.e. $\|A\|_F/n \leq \|B\|_F/m$ (with $r$ in denominators canceled). The design of stable condition is motivated by the terms $\delta_1 = s\eta g_B^t A_t Z$ and $\delta_2 = s\eta B_t g_A^t Z$, which reach similar scales when $A_t$ and $B_t$ do. Since $\gamma[\delta_2] = 0$ is always ensured, satisfying this condition also guarantees $\gamma[\delta_1] = 0$. The practical effectiveness of this stable condition has been demonstrated in Section 5.3.

Stable-LoRA can robustly prevent instable feature learning for any learning rate $\eta$: after $N$ steps of shrinking, we have

$$
\begin{aligned}
A_N &= (1 - \lambda)A_{N-1} - \eta g_A^{N-1} \\
&= (1 - \lambda)^2 A_{N-2} - (1 - \lambda)\eta g_A^{N-2} - \eta g_A^{N-1} \\
&= \cdots \\
&= (1 - \lambda)^N A_0 - \eta g_A^{N-1} - \eta((1 - \lambda)g_A^{N-2} + \cdots \\
&\quad + (1 - \lambda)^{N-1}g_A^0) \\
&= (1 - \lambda)^N A_0 - \eta g_A^{N-1} - \eta\Delta
\end{aligned}
$$

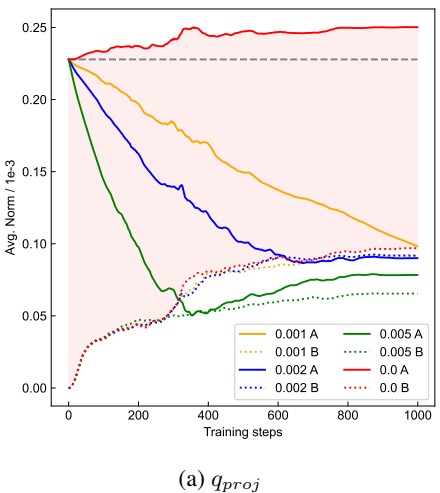
(a) $q_{proj}$

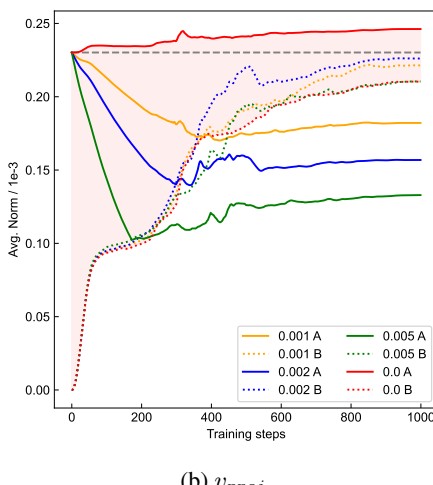
(b) $v_{proj}$

Figure 2: Averaged norm of $A$ and $B$ on the 0.5B model and HellaSwag. While the scale of $B$ is smaller, it grows more rapidly than $A$ ($|g_B| > |g_A|$), indicating a practical violation of feature learning stability.

With $\gamma[(1-\lambda)^k] \leq \gamma[1] = 0$ for all $k \in \mathbb{Z}^+$, we have $\gamma[\Delta Z] \leq \gamma[g_A^* Z] = 1$. Therefore,

$$\gamma[A_N Z] = \max(N\gamma[1-\lambda] + \gamma[A_0 Z], \gamma[\eta] + \gamma[g_A^{N-1} Z], \gamma[\eta] + \gamma[\Delta Z])$$
$$= \max(N\gamma[1-\lambda] + \gamma[A_0 Z], \gamma[\eta] + 1)$$

With $N$ and/or $\lambda$ sufficiently large, $N\gamma[1-\lambda] + \gamma[A_0 Z]$ will finally drop below $\gamma[\eta] + 1$, and hence stable feature learning is achieved from step $N + 1$ onward and persists throughout the rest of training.

Stable-LoRA is orthogonal to existing optimization strategies such as gradient optimization (like AdamW) and weight decay, as formally described in Algorithm 1. More importantly, Stable-LoRA incurs negligible overheads during training: it requires no additional memory usage, since the shrinkage operation can be done in-place (and should be, for acceleration) with the pre-shrinkage value no longer used after that. This property is particularly crucial since LoRA is commonly used in memory-constrained scenarios. Computation overhead arises from computing the Frobenius norms $\|\cdot\|_F$ and performing the shrinkage, which is negligible (as shown in Table 5) because (1) the operations are relatively light-weight compared to others, and (2) they are applied only during the initial steps.

## 5 EXPERIMENTS

We evaluated Stable-LoRA and other baselines under these experimental settings:

**Datasets**. The tasks involve two fine-tuning scenarios: multi-choice question answering (QA) and chain-of-thought (CoT) reasoning. The QA datasets include HellaSwag (Zellers et al., 2019), SocialIQa (Sap et al., 2019), OpenbookQA (Mihaylov et al., 2018), ARC-Easy and ARC-Challenge (Clark et al., 2018). For CoT reasoning, we focus on mathematical tasks where models are trained on MetaMathQA (Yu et al., 2023) and evaluated on GSM8K (Cobbe et al., 2021). The exact match accuracy is used as evaluation metric for all tasks.

**Models**. The experiments are conducted on the 0.5B and 1.5B models from Qwen-2 (Yang et al., 2024) and 1B and 3B models from LLaMA-3.2 (Dubey et al., 2024).

**Baselines**. Besides AdamW, we compared our proposed method against several other baselines, including stable feature learning methods of LoRA+ (Hayou et al., 2024b) and Riemann Preconditioned Optimization (Zhang & Pilanci, 2024), and a transformation-invariant optimizer LoRA-RITE (Yen et al., 2025). LoRA+ claims to achieve stable feature learning by setting learning rate of $B$

| Model | Method | HellaS. | SIQA | ObQA | ARC-E | ARC-C | Avg. |
|---|---|---|---|---|---|---|---|
| 0.5B | AdamW | 65.65 | 67.04 | 63.00 | 66.92 | 47.10 | 61.94 |
| | LoRA+ | 64.75 | 67.91 | 64.40 | 67.80 | 47.87 | 62.55 |
| | Riemann | 60.79 | 66.94 | 65.00 | 67.30 | 47.35 | 61.48 |
| | LoRA-RITE | 62.73 | 66.89 | 66.00 | 67.13 | 46.50 | 61.85 |
| | **Stable-LoRA** | **66.73** | **68.27** | **67.00** | **69.15** | **48.89** | **64.01** |
| 1B | AdamW | 83.76 | 71.70 | 73.20 | 76.81 | 54.01 | 71.90 |
| | LoRA+ | 83.39 | 71.19 | 73.20 | 75.88 | 53.16 | 71.36 |
| | Riemann | 77.41 | 70.62 | 71.20 | 73.99 | 52.13 | 69.07 |
| | LoRA-RITE | 82.38 | 71.60 | 71.00 | 76.73 | 53.50 | 71.04 |
| | **Stable-LoRA** | **84.41** | **72.26** | **73.80** | **77.36** | **54.78** | **72.52** |
| 1.5B | AdamW | 88.28 | 77.33 | 83.60 | 86.20 | 70.14 | 81.11 |
| | LoRA+ | 87.94 | 77.38 | 83.20 | 85.94 | 70.65 | 81.02 |
| | Riemann | 86.46 | 76.87 | 82.40 | 85.82 | 69.71 | 80.25 |
| | LoRA-RITE | 87.97 | 77.23 | 83.00 | 86.36 | 70.82 | 81.08 |
| | **Stable-LoRA** | **88.52** | **77.64** | **84.00** | **86.99** | **72.61** | **81.95** |
| 3B | AdamW | 93.39 | 79.89 | 83.20 | 88.38 | 72.78 | 83.53 |
| | LoRA+ | 93.54 | 79.94 | 83.00 | 88.26 | 72.35 | 83.42 |
| | Riemann | 92.61 | 79.89 | 82.60 | 88.01 | 71.42 | 82.91 |
| | LoRA-RITE | 93.21 | 80.25 | 83.40 | 88.26 | 71.50 | 83.32 |
| | **Stable-LoRA** | **93.59** | **80.25** | **84.00** | **88.68** | **73.63** | **84.03** |

Table 1: Task accuracies of models on datasets of question-answering tasks.

larger than $A$. Riemann Preconditioned Optimization adopts matrix preconditioning on $g_A$ and $g_B$. LoRA-RITE achieves invariant transformation equilibration of LoRA using unmagnified gradients.

**Configurations**. Unless otherwise stated, we use the following training configurations. We train $q_{proj}$ and $v_{proj}$ of the attention block with rank $r = 8$. We conducted careful tuning of hyper-parameters by searching $\eta$ from 5e-5 to 8e-4 and $s$ from 2.0 to 64.0. Each value of $A_0$ is sampled from $[-1/n, 1/n]$ following (He et al., 2015) and $B_0 = 0$. For LoRA+, we set $\eta_B = 4\eta_A$ following its recommendation for decoder-only models. We search the shrinkage ratio over $\lambda \in [0.0005, 0.001, 0.002, 0.005]$ and report the best result (more detailed results about each value of $\lambda$ are in Table 8). The algorithm of AdamW is adopted as the gradient optimizer for Stable-LoRA. Each reported accuracy is the best of 3 random runs. More detailed configurations are specified in corresponding sections or Table 7.

## 5.1 Dynamic analysis

Figure 2 provides a dynamic analysis of the training process by visualizing the change of Frobenius norms $||\cdot||_F$ of matrices $A$ and $B$. The results indicate that **the problem of $\gamma[A_t Z] > \gamma[\eta] + 1$ does occur in practice**:

During LoRA training, $||B||_F$ grows rapidly from small values, meaning that $g_B$ is large while the value of $B$ is small. Meanwhile, $||A||_F$ remains at a steady but larger value, meaning that $g_A$ small and $A$ large. Recall that $\delta_1 = s\eta g_B^t A_t Z$ and $\delta_2 = s\eta B_t g_A^t Z$, the larger $g_B$ and $A$ makes $\delta_1$ dominate over $\delta_2$ ($\gamma[\delta_1] > \gamma[\delta_2]$). This is consistent with our theoretical analysis that $\gamma[A_t Z] > \gamma[\eta] + 1$. Furthermore, $||A||_F$ never drops below its initial value, indicating that the negative influence of initialization persists throughout training.

Stable-LoRA effectively mitigates this negative effect and hence promotes stable feature learning. Moreover, the value of $||B||_F$ in early training stages remains unaffected when Stable-LoRA declines $||A||_F$, confirming that our method preserves the benefit of non-zero $A_0$.

| Method | 1B | | | 3B | | |
|---|---|---|---|---|---|---|
| | 1000 | 2000 | 5000 | 1000 | 2000 | 5000 |
| AdamW | 23.88 | 27.14 | 31.08 | 51.48 | 53.45 | 58.83 |
| **Stable-LoRA** | **24.56** | **27.75** | **31.84** | **52.16** | **54.74** | **59.44** |

Table 2: Task accuracies of models on (math) CoT reasoning tasks for different training steps.

| Model | Target | | | | | |
|---|---|---|---|---|---|---|
| | *qv* | | *qkvo* | | *qkvogud* | |
| | AdamW | Stable-LoRA | AdamW | Stable-LoRA | AdamW | Stable-LoRA |
| 0.5B | 61.54 | **62.46** | 63.03 | **63.48** | 64.35 | **64.75** |
| 1B | 71.90 | **72.80** | 73.76 | **73.96** | 75.04 | **75.44** |
| 1.5B | 80.27 | **80.75** | 80.94 | **81.15** | 81.71 | **81.82** |
| 3B | 83.79 | **84.13** | 84.54 | **84.93** | 85.31 | **85.60** |

Table 3: Task accuracies of models on the combined dataset of question-answering tasks.

## 5.2 MAIN RESULTS

### 5.2.1 RESULTS OF MULTI-CHOICE QUESTION ANSWERING

Table 1 presents the results on the QA datasets. As demonstrated, Stable-LoRA consistently outperforms other methods across models and datasets, achieving up to a 4% increase in accuracy. While other baselines may boost performances on specific tasks or models, the improvements are inconsistent. In contrast, Stable-LoRA offers not only improved accuracies but also greater robustness across tasks and models.

### 5.2.2 RESULTS OF CHAIN-OF-THOUGHT REASONING

Chain-of-Thought (CoT) is a widely used approach in tasks that require multi-step reasoning. We trained models to learn to reason in CoT format spontaneously without explicit prompting (i.e., giving the question directly without instructing the model to "think step by step"). We used math-reasoning datasets as representative reasoning tasks. The results in Table 2 show that Stable-LoRA again outperforms AdamW, maintaining its performance advantages in CoT tasks.

### 5.2.3 ABLATIONS

To evaluate the generalization capability of our method, we perform ablation studies along two dimensions: dataset formulation and target modules. For the dataset formulation study, we construct a unified QA dataset by combining all datasets (see details in Section C.1). We also vary the target matrix components of the model to which LoRA is applied. As shown in Table 3, Stable-LoRA consistently improves task accuracy across different LoRA configuration settings. Additional ablation results are provided in Section D.3.

## 5.3 MEMORY AND COMPUTATIONAL COSTS.

Stable-LoRA introduces no additional memory usage compared to LoRA, as the shrinkage operation is conducted in-place. Table 5 compares the training time of Stable-LoRA with baseline methods. The results show that Stable-LoRA incurs only a marginal (0.6%) increase in training time, indicating that the scalar-matrix multiplication involved in shrinkage is far less costly than gradient computation and parameter updates.

| Method | 0.5B | 1B | 1.5B | 3B |
|---|---|---|---|---|
| Stable-LoRA | 62.46 | **72.80** | **80.75** | **84.13** |
| Stable-LoRA w/o the stable condition | **62.52** | 72.72 | 80.61 | 84.10 |

Table 4: Task accuracies on the combined QA dataset from Stable-LoRA with/without the stable condition.

| Method | AdamW | LoRA+ | Riemann | LoRA-RITE | Stable-LoRA |
|---|---|---|---|---|---|
| Time(s) | 217.4 | 217.4 | 235.5 | 317.3 | 218.8 |
| +% | - | +0.0% | +8.3% | +46.0% | +0.6% |

Table 5: Comparison of training time of different methods on 0.5B model and HellaSwag.

## 5.4 JUSTIFICATION FOR THE STABLE CONDITION.

To justify the stable condition, we conducted experiments where the stable condition is removed and $A$ shrinks whenever $\|A\|_F/n > \|B\|_F/m$. Table 4 compares task accuracies with and without stopping at the stable condition, and the results show that further shrinkage beyond the stable condition does not lead to noticeable impact on performance, which aligns with our previous analysis.

## 6 RELATED WORKS

### 6.1 STABLE FEATURE LEARNING.

There are existing works established upon initialization schemes for stable feature learning, from the perspective of width and depth. In scenario of width, (Glorot & Bengio, 2010) proposed Xavier initialization to stabilize the variance of activations, and (He et al., 2015) improved it for non-linear activation functions (like leaky ReLu). (Yang & Hu, 2021) introduced $\mu P$ parameterization for ensuring feature learning in the infinite-width scenario. Related literature about the depth limit includes (Hayou, 2022; Schoenholz et al., 2016) etc.. Stable-LoRA is specifically targeted at width scenario, so it is theoretically discussed upon the width-related initialization method of (He et al., 2015) and shows empirically strong results.

### 6.2 STABLE FEATURE LEARNING FOR LORA.

The concept of LoRA stable feature learning originates from LoRA+ (Hayou et al., 2024b), which suggests choosing a larger learning rate for $B$ than $A$ ($\eta_B > \eta_A$). (Zhang & Pilanci, 2024) proposed a matrix-preconditioned optimizer to achieve stabilization. Beyond width-related stability, (Kalajdzievski, 2023) studies the stability with respect to rank $r$ and recommends using scaling factor $s = \alpha/\sqrt{r}$ rather than $\alpha/r$. Our definition of stable feature learning is slightly different from the above-mentioned work, where we do not demand intermediate states to be $\Theta(n^0)$ due to practical considerations.

## 7 CONCLUSION

This paper addresses the challenge of stabilizing feature learning in Low-Rank Adaptation (LoRA). We first establish that, under appropriate hyper-parameters and initializations of $A$ and $B$, LoRA can in principle be self-stabilized during the training process regardless of model width, which provides a theoretical foundation for the robustness and effectiveness of LoRA. However, we further reveal that the non-zero $A_0$ compromises this self-stability which leads to performance degradation. Stable-LoRA is hence proposed as a weight-shrinkage strategy that mitigates instability caused by $A_0$ while preserving its benefits. Stable-LoRA shows superiority over various tasks and models, with no additional memory usage and only marginal computation overhead.

ACKNOWLEDGEMENT

This work is partially supported by the NSF of China (under Grant 92364202), and Major Program of ISCAS (Grant No. ISCAS-ZD-202402).

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

## A    RELATED WORKS OF LoRA VARIANTS.

Besides LoRA, there are some similar strategies of low-rank adapters proposed. DoRA (Liu et al., 2024) seperately updates the normalized matrices and its norm, leading to more similar behavior to Full Fine-Tuning. VeRA (Kopiczko et al., 2024) creates a random and frozen matrix and learns vector scalings of the columns, to achieve extreme memory saving. HiRA (Huang et al., 2025) element-wise product of $W$ and $BA$ to enlarge the rank of updates. QLoRA (Dettmers et al., 2023) reduces computation costs by quantizing pretrained weights down to smaller bits, and only set higher bits for trainable parameters, to save memory usage. Flora (Hao et al., 2024) achieves overall high-rank updates by resampling low-rank projection matrices for each step and accumulating them periodically.

## B    DETAILS OF ALGORITHM

Algorithm 1 shows the detailed procedure of Stable-LoRA, where the orthogonality of our method to gradient optimizers and weight decay is demonstrated.

Note that Stable-LoRA is conceptually different from weight decay from several perspectives:

- **Theoretical motivation and formulation.** Weight decay is based on a Bayesian prior assuming that model weights follow a Gaussian distribution centered at zero. It introduces an additional regularization term in the loss function, so the rate $w$ is multiplied by $\eta$ and then applied to the parameters.

$$A = A - \eta w A = (1 - \eta w)A$$

In contrast, Stable-LoRA directly shrinks the weights using $\lambda$ which is independent of the learning rate:

$$A = A - \lambda A = (1 - \lambda)A$$

With $\eta << 1$ in almost all cases, Stable-LoRA results in significantly faster decay compared to weight decay.

- **Scope of application.** Weight decay is applied uniformly to all trainable parameters, and Stable-LoRA targets only at the matrix $A$, with the explicit purpose of reducing the influence of $A_0$.

- **Application schedule.** Weight decay is applied throughout the entire training process, while Stable-LoRA is only applied before the stable condition achieved.

## C    DETAILS OF EXPERIMENTS

### C.1    DATASETS

Table 6 contains the detailed information of datasets. All datasets include test sets but no validation sets; therefore we apply weight decay to mitigate overfitting. For the QA datasets, we use the templated versions provided by (Hu et al., 2023).

The combined QA dataset includes 8K samples each from HellaSwag and SIQA, as well as all available samples from ObQA, ARC-E, and ARC-C, resulting in approximately 24K samples in total.

### C.2    HYPER-PARAMETERS

The hyper-parameters of experiments (except for specified ones) are listed in Table 7. To find the optimal combination, $\eta$ and $s$ are thoroughly searched across wide ranges. The weight decay ratio $w$ is set to 0.01 to prevent overfitting, and the dropout ratio is 0.0 as setting a non-zero value of it cannot boost performance in our experiments.

We set the training steps to 1,000 for each individual QA dataset and 3,000 for the combined QA dataset, ensuring that the effective number of training epochs per QA task remains consistent between the separate and combined settings.

| Dataset | #Train | #Test |
|---------|--------|-------|
| HellaSwag | 39905 | 10042 |
| SIQA | 33410 | 1954 |
| ObQA | 4957 | 500 |
| ARC-E | 2251 | 2376 |
| ARC-C | 1119 | 1172 |
| MetaMathQA | 40000 | 1319 |

Table 6: Detailed information about datasets.

| Hyper-parameter | Value |
|-----------------|-------|
| Learning rate $\eta$ | [5e-5,$\{1\sim8\}$e-4] |
| Scale factor $s$ | 2.0 to 64.0 (interval 1.0) |
| Target modules | $q_{proj}, v_{proj}$ |
| Rank $r$ | 8 |
| Weight decay ratio $w$ | 0.01 |
| Dropout ratio | 0.0 |
| Batch size | 16 |
| Learning rate scheduler | linear |
| Warm-up steps | 100 |
| AdamW $\beta_1, \beta_2$ | 0.9, 0.999 |

Table 7: Hyper-parameters.

# D  ADDITIONAL RESULTS

## D.1  RESULTS OF DIFFERENT $\lambda$S

We show results of different $\lambda$s on QA datasets in Table 8. The results show that Stable-LoRA can robustly outperform AdamW with a variety of $\lambda$, demonstrating the principle effectiveness of the method. $\lambda$ should be considered as a newly-introduced hyper-parameter, which could and should be tuned for optimal performances. The results of $\lambda = 0.005$ is often suboptimal, suggesting that an excessively larger $\lambda$ could result in loss of information from previous training steps and hence downgraded performances.

## D.2  RESULTS ON LARGER SCALE

Table 9 shows results of Llama-3.1-8B with targets of $qkvogud$ on the math-reasoning datasets, which verifies performance superiority of Stable-LoRA on the larger-scaled model.

## D.3  RESULTS OF MORE ABLATIONS

Table 10 shows the results of ablation studies on different models, target modules and datasets, where Stable-LoRA shows uniform superiority of performances across the involved tasks.

## D.4  COMPARISON WITH LORA VARIANTS

Table 11 reports the performance of Stable-LoRA and DoRA on the QA benchmarks. The results show that Stable-LoRA empirically outperforms DoRA across these tasks.

We note that Stable-LoRA cannot be directly applied to substantially different variants such as DoRA, since the variants fundamentally modifies the original LoRA parameterization (e.g.DoRA changes it to $m\frac{W_0+BA}{||W_0+BA||_c}$), resulting in distinct training dynamics.

| Target | Method | 0.5B | 1B | 1.5B | 3B |
|--------|--------|------|----|----|----|
| | AdamW | 61.54 | 71.90 | 80.27 | 83.79 |
| | Stable-LoRA$_{\lambda=0.0005}$ | 62.36 | **72.80** | **80.75** | **84.13** |
| $qv$ | Stable-LoRA$_{\lambda=0.001}$ | 62.13 | 72.37 | 80.55 | 84.08 |
| | Stable-LoRA$_{\lambda=0.002}$ | 62.05 | 71.92 | 80.44 | 83.71 |
| | Stable-LoRA$_{\lambda=0.005}$ | **62.46** | 72.29 | 80.50 | 83.85 |
| | AdamW | 63.03 | 73.76 | 80.94 | 84.54 |
| | Stable-LoRA$_{\lambda=0.0005}$ | 63.21 | **73.96** | 81.09 | 84.80 |
| $qkvo$ | Stable-LoRA$_{\lambda=0.001}$ | 63.15 | 73.54 | 81.12 | 84.85 |
| | Stable-LoRA$_{\lambda=0.002}$ | **63.48** | 73.84 | **81.15** | **84.93** |
| | Stable-LoRA$_{\lambda=0.005}$ | 63.09 | 73.78 | 80.89 | 84.53 |
| | AdamW | 64.35 | 75.04 | 81.71 | 85.31 |
| | Stable-LoRA$_{\lambda=0.0005}$ | 64.37 | 75.42 | 81.71 | **85.60** |
| $qkvogud$ | Stable-LoRA$_{\lambda=0.001}$ | 64.53 | **75.44** | **81.82** | 85.32 |
| | Stable-LoRA$_{\lambda=0.002}$ | **64.75** | 75.20 | 81.74 | 85.56 |
| | Stable-LoRA$_{\lambda=0.005}$ | 64.33 | 75.18 | 81.71 | 85.23 |

Table 8: Task accuracies of different $\lambda$s on the combined QA datasets.

| Step | AdamW | Stable-LoRA$_{\lambda=0.0005}$ | Stable-LoRA$_{\lambda=0.001}$ | Stable-LoRA$_{\lambda=0.002}$ |
|------|-------|------|------|------|
| 1000 | 68.61 | 69.75 | **70.05** | 69.60 |
| 2000 | 70.96 | 71.42 | 71.49 | **71.87** |
| 3000 | 71.72 | **74.37** | 73.39 | 72.02 |

Table 9: Task accuracies of Llama-3.1-8B with targets of $qkvogud$ on the math reasoning dataset.

# E  LIMITATIONS

The effectiveness of Stable-LoRA has been validated in two LLM fine-tuning scenarios. However, its performance in other settings—such as additional datasets or cross-modal tasks (e.g., vision)—remains to be explored. We employ AdamW as the gradient optimizer, and compatibility with alternative optimizers has not yet been systematically evaluated. For initialization, we adopt the commonly used scheme $A_0 \sim [-1/n, 1/n]$ and $B_0 = 0$, rather than alternative strategies.

| Model | Target | Method | HellaS. | SIQA | ObQA | ARC-E | ARC-C | Avg. |
|-------|--------|--------|---------|------|------|-------|-------|------|
| 0.5B | *qkvo* | AdamW | 68.00 | 68.42 | 67.20 | 68.73 | 48.98 | 64.27 |
| | | Stable-LoRA | 68.98 | 68.83 | 67.80 | 69.57 | 49.83 | 65.00 |
| | *qkvogud* | AdamW | 68.76 | 69.14 | 67.60 | 69.49 | 48.04 | 64.61 |
| | | Stable-LoRA | 69.08 | 69.50 | 68.80 | 70.79 | 49.74 | 65.58 |
| 1B | *qkvo* | AdamW | 85.09 | 73.44 | 74.60 | 77.23 | 55.20 | 73.11 |
| | | Stable-LoRA | 86.21 | 73.90 | 74.60 | 78.16 | 56.51 | 73.88 |
| | *qkvogud* | AdamW | 85.93 | 74.36 | 76.80 | 77.44 | 55.03 | 73.91 |
| | | Stable-LoRA | 86.72 | 74.72 | 77.40 | 78.58 | 57.17 | 74.92 |
| 1.5B | *qkvo* | AdamW | 89.13 | 77.69 | 84.20 | 86.91 | 70.56 | 81.70 |
| | | Stable-LoRA | 89.32 | 78.05 | 85.00 | 87.63 | 72.35 | 82.47 |
| | *qkvogud* | AdamW | 89.72 | 78.66 | 84.80 | 87.12 | 71.25 | 82.31 |
| | | Stable-LoRA | 89.83 | 78.66 | 85.80 | 87.79 | 72.35 | 82.89 |
| 3B | *qkvo* | AdamW | 94.02 | 80.45 | 83.60 | 88.43 | 72.44 | 83.79 |
| | | Stable-LoRA | 94.23 | 80.76 | 85.00 | 88.89 | 74.06 | 84.59 |
| | *qkvogud* | AdamW | 94.41 | 80.81 | 84.80 | 88.51 | 72.61 | 84.23 |
| | | Stable-LoRA | 94.50 | 81.01 | 85.60 | 88.72 | 74.23 | 84.81 |

Table 10: Task accuracies of models and different targets on QA datasets. The top and bottom value are from AdamW and Stable-LoRA respectively.

| Model | Method | HellaS. | SIQA | ObQA | ARC-E | ARC-C | Avg. |
|-------|--------|---------|------|------|-------|-------|------|
| 1B | DoRA | 82.47 | 71.49 | 73.60 | 76.98 | 53.84 | 71.68 |
| | Stable-LoRA | **84.41** | **72.26** | **73.80** | **77.36** | **54.78** | **72.52** |
| 3B | DoRA | 93.38 | 80.04 | 82.80 | 88.47 | 72.27 | 83.39 |
| | Stable-LoRA | **93.59** | **80.25** | **84.00** | **88.68** | **73.63** | **84.03** |

Table 11: Task accuracies of DoRA and Stable-LoRA on 1B and 3B models, $qv$ and QA datasets.

