# OpenReview forum: "Stable-LoRA: Stabilizing Feature Learning of Low-Rank Adaptation"
_ICLR.cc/2026/Conference — ICLR 2026 Poster_

### Official Review · Reviewer_iYHY · 2025-10-31

**Soundness:** 2
**Presentation:** 3
**Contribution:** 3
**Rating:** 4
**Confidence:** 3

**Summary:**

This paper provides a theoretical analysis of LoRA’s fine-tuning dynamics and argues that LoRA can achieve self-stabilization under proper initialization and hyperparameter choices. The authors further propose *Stable-LoRA*, a simple weight-shrinkage strategy applied to the A matrix during early training to mitigate instability. Experimental results show consistent improvements across several reasoning and QA benchmarks, demonstrating the method’s effectiveness and efficiency.

**Strengths:**

The paper offers a clear theoretical perspective on the stability of LoRA fine-tuning, supported by a consistent use of the γ-function framework to characterize scaling behavior. The derivations are mathematically well-structured and connect intuitively to optimization dynamics. The proposed Stable-LoRA method is simple yet effective, introducing negligible computational overhead while improving training stability across tasks. The work also contributes conceptually by bridging theoretical analysis and practical optimization design within the PEFT paradigm.

**Weaknesses:**

1. About Assumption 1:

   If the activation $Z$ is normalized under a fan-in scaling scheme  (i.e., each input element scaled by $1/\sqrt{n}$, which is common in linear layers or attention projections),  then each component of $Z$ becomes $\Theta(n^{-1/2})$.

   Consequently, the product $g_A^t Z$ involves a summation over $n$ such terms,  resulting in an overall magnitude of $\Theta(\sqrt{n})$.  This implies that $\gamma[g_A^t Z] = \tfrac{1}{2}$ rather than $1$.  Could the authors clarify whether Assumption 1 still holds under common fan-in scaling conventions?  In particular, does the assumed normalization (e.g., layer normalization, residual connections, or other schemes)  preserve the activations at the $\Theta(1)$ scale?

2. In Section 3.2, the authors assume that $\gamma[A_0 Z] \le \gamma[\eta] + 1 \Rightarrow \gamma[A_t Z] = \gamma[\eta] + 1.$ However, “≤” only provides an upper bound and does not guarantee equality during training. This step effectively assumes convergence of the recursive relation $\gamma[A_t Z] = \max(\gamma[A_{t-1}Z], \gamma[\eta] + 1),$ without proving that the sequence will reach equality. Could the authors provide a justification (analytical or empirical) showing that the recursion indeed converges to $\gamma[A_t Z] = \gamma[\eta] + 1$ rather than remaining strictly below that bound?

3. The paper claims that when only one condition in Eq. (5) holds, $\gamma[A_t Z] = \gamma[B_t] + 1 \Rightarrow \gamma[\delta_1] = \gamma[\delta_2],$ which is said to be “undesirable.” However, δ₁ and δ₂ represent the contributions of matrices A and B to the output update ΔYₜ. Having the same γ-scale simply implies that both pathways contribute symmetrically, which is not necessarily detrimental to learning stability. Could the authors clarify why $\gamma[\delta_1] = \gamma[\delta_2]$ must be considered suboptimal? Is there theoretical or empirical evidence that symmetric scaling between A and B leads to instability?

### Cons

1. The baselines considered in Table 1 are too limited. Recent PEFT methods such as DoRA[1] and AdaLoRA  also address optimization stability and efficiency. Including these methods would provide a fairer and more comprehensive comparison.

2.  The backbones used are relatively small (≤ 3B parameters). To substantiate the claim of general stability, experiments on larger or more recent architectures (e.g., Llama-3.2-8B, Mistral-7B, or Mixtral-8×7B) are necessary. This would also help assess scalability and compatibility with modern large-model training dynamics.

   [1] Liu, X., Li, Y., Zhou, T., Wang, K., & Qiu, X. (2024). *DoRA: Weight-Decomposed Low-Rank Adaptation*. arXiv preprint arXiv:2402.09353.

   [2] Zhang, R., Xu, H., Cui, Y., Liu, T., & Zhang, Y. (2023). *AdaLoRA: Adaptive Budget Allocation for Parameter-Efficient Fine-Tuning*. arXiv preprint arXiv:2303.10512.

## Typos

1. Lines 101: There is a dot(.) at the begining of 101 lines. Maybe its should be placed on the end of Eq. (1).

2. Lines 202: The same issue as in line 101 — punctuation should not appear at the beginning of a new line.

**Questions:**

Please see the weaknesses.

---

> ### Author Response · Authors · 2025-11-21
>
> We sincerely thank you for the review!
>
> > If the activation $Z$ is normalized under a fan-in scaling scheme (i.e., each input element scaled by $1/\sqrt{n}$, which is common in linear layers or attention projections), then each component of $Z$ becomes $\Theta(n^{-1/2})$. Consequently, the product $g_A^t Z$ involves a summation over such terms, resulting in an overall magnitude of $\Theta(\sqrt{n})$. This implies that $\gamma[g_A^t Z] = \tfrac{1}{2}$ rather than $1$. Could the authors clarify whether Assumption 1 still holds under common fan-in scaling conventions? In particular, does the assumed normalization (e.g., layer normalization, residual connections, or other schemes) preserve the activations at the $\Theta(1)$ scale?
>
> 1.
>
> (1) If the activation $Z$ is a uniformed vector (all the items are equal), then after layernorm each item of $Z$ would be $1/\sqrt{n}$. However, in real scenario the value of $Z$ is not uniformed, and the components of layer-normed results may not be evenly $\Theta(n^{-1/2})$.
>
> (2) It should also be noted that $\gamma[g_A^t Z] \ne \gamma[g_A^t] + \gamma[Z]$, as $g_A^tZ$ involves $n$-item addition in the matmul ( $g_A^t \in R^{r\times n}$ , $Z \in R^{n\times (bsz \times len)}$ ). When $n$ is involved in computation, the multiplication law of $\gamma$ no longer fits. Here is a simpler example: if we multiply $1^{1\times n}$ with $1^{n\times 1}$, we will get a scalar $n$ with $\gamma[n]=1$, rather than $\gamma[1^{1\times n}] + \gamma[1^{n\times 1}]=0+0=0$. You can see from Appendix B that $\gamma[g_A^t Z]=1$ is also induced by $n$-item addition in the matmul. Consequently, we cannot conclude $\gamma[g_A^tZ]=\tfrac{1}{2}$ from $\gamma[g_A^t]=1$ and $\gamma[Z]=\tfrac{1}{2}$.
>
> 2. Assumption 1 has been widely adopted and clarified in prior works [1-3], as a foundation of theoretical analysis of stable feature learning.
>
> 3. Only the modules that are directly after layernorm (e.g. q,k,v) will receive a layer normalized vector as input. The input to other modules, however, is never layer normalized. For example, the input to o\_proj is computed from q,k and v and therefore not normalized.
>
> 4. Even if you substitute all the $1$ by $1/2$, you will find out that all the claims still hold, including Theorem 3.1 and all its induction. The cause of instability largely lies in that $\gamma[A_0Z]$ is too larger compared to $\gamma[\eta]$, rather than $1$.
>
> > In Section 3.2, the authors assume that $\gamma[A_0 Z] \le \gamma[\eta] + 1 \Rightarrow \gamma[A_t Z] = \gamma[\eta] + 1$. However, '$\leq$' only provides an upper bound and does not guarantee equality during training. This step effectively assumes convergence of the recursive relation $\gamma[A_t Z] = \max(\gamma[A_{t-1}Z], \gamma[\eta] + 1)$ without proving that the sequence will reach equality. Could the authors provide a justification (analytical or empirical) showing that the recursion indeed converges to $\gamma[A_t Z] = \gamma[\eta] + 1$ rather than remaining strictly below that bound?
>
> Due to the max operator in $\gamma[A_t Z] = \max(\gamma[A_{t-1}Z], \gamma[\eta] + 1)$, it is quite straight forward that $\gamma[A_t Z] = \gamma[\eta] + 1$ if $\gamma[A_0 Z] \le \gamma[\eta] + 1$. The latter term $\gamma[\eta] + 1$ is clearly the larger value in the expression of $\max(\gamma[A_{t-1}Z], \gamma[\eta] + 1)$.
>
> > The paper claims that when only one condition in Eq. (5) holds, $\gamma[A_t Z] = \gamma[B_t] + 1 \Rightarrow \gamma[\delta_1] = \gamma[\delta_2]$, which is said to be 'undesirable'. However, δ₁ and δ₂ represent the contributions of matrices A and B to the output update ΔYₜ. Having the same γ-scale simply implies that both pathways contribute symmetrically, which is not necessarily detrimental to learning stability. Could the authors clarify why $\gamma[\delta_1] = \gamma[\delta_2]$ must be considered suboptimal? Is there theoretical or empirical evidence that symmetric scaling between A and B leads to instability?
>
> Actually, our claim is indeed the opposite: $\gamma[\delta_1] = \gamma[\delta_2]$ is a good condition, instead of bad. Line 165-169 clearly explained the reason that imbalanced scaling is equivalent to fixing $A$ or $B$ as $n$ scales, which is definitely a suboptimal case. That is also the reason why both conditions in Eq. (5) must simultaneously hold or not, for achieving $\gamma[\delta_1] = \gamma[\delta_2]$. Similar claims can also be found in LoRA+[1].

---

> ### Author Response · Authors · 2025-11-21
>
> > The baselines considered in Table 1 are too limited. Recent PEFT methods such as DoRA and AdaLoRA also address optimization stability and efficiency. Including these methods would provide a fairer and more comprehensive comparison.
>
> As far as we know, all the feature-learning related works have been included in our paper as baselines. Following your suggestion, we conducted extensive experiments on DoRA, which is a structural variant of LoRA, and the results are in the table below. AdaLoRA targets at rank allocation rather than optimization, so we did not experiment on it.
>
> |  | DoRA | Stable-LoRA |
> | --- | --- | --- |
> | Llama-3.2-1B |  |  |
> | Hellas.   | 82.47   | 83.80   |
> | SIQA   |  71.17  | 71.92   |
> | OBQA   |  72.07  |  72.80  |
> | ARC-e   |  76.42  |  77.20  |
> | ARC-c   | 53.16   |  54.44  |
> | Llama-3.2-3B |  |  |
> | Hellas.   |  93.30  |  93.51  |
> | SIQA   |  79.58  | 80.07   |
> | OBQA   |  82.40  | 83.60   |
> | ARC-e   |  88.25  |  88.29  |
> | ARC-c   |  72.04  |  73.04  |
>
> As shown, Stable-LoRA shows outperforming performances against DoRA across model types and tasks.
>
>
> > The backbones used are relatively small (≤ 3B parameters). To substantiate the claim of general stability, experiments on larger or more recent architectures (e.g., Llama-3.2-8B, Mistral-7B, or Mixtral-8×7B) are necessary. This would also help assess scalability and compatibility with modern large-model training dynamics.
>
> We conducted additional experiments on Llama-3.1-8B and the reasoning task (the 3.2 version has only 1B and 3B models). Following the suggestion, target modules are set to all linear weights for the largest scale. The results are in the table below, which demonstrates that Stable-LoRA can also achieve ourperforming performances on the larger-scale configuration.
>
> | AdamW | Stable-LoRA $_{\lambda=0.0005}$ | Stable-LoRA $_{\lambda=0.001}$ | Stable-LoRA $_{\lambda=0.002}$ |
> | --- | --- | --- | --- |
> | 68.61 | 69.75 | 70.05 | 69.60 |
>
>
> Lastly, we appreciate that you pointed out the typos, and we will fix them in the new version.
>
> [1]Lora+:Efficient low rank adaptation of large models.In International Conferenceon Machine Learning, pp. 17783–17806. PMLR, 2024.
>
> [2]Riemannian preconditioned lora for fine-tuning foundation models.In Proceedings of the 41st International Conference on Machine Learning, pp. 59641–59669, 2024
>
> [3]The impact of initialization on lora finetuning dynamics. Advances in Neural Information Processing Systems, 37:117015–117040, 2024.

---

### Official Review · Reviewer_S7AS · 2025-10-31

**Soundness:** 3
**Presentation:** 3
**Contribution:** 3
**Rating:** 6
**Confidence:** 4

**Summary:**

This paper introduces Stable-LoRA, a novel optimization strategy that enhances the training stability of LoRA. The authors first theoretically demonstrate that LoRA possesses self-stabilizing properties under appropriate hyperparameters and initialization, but commonly used non-zero initialization often undermines this stability. To address this, Stable-LoRA dynamically shrinks matrix A during the early training phase. Experimental results demonstrate the effectiveness of Stable-LoRA and with negligible additional computational or memory overhead.

**Strengths:**

* This paper is solid in its theoretical contribution. The motivation and concepts are well illustrated, making the work easy to follow. The algorithm design is simple yet elegant, and the stability stopping criterion is theoretically justified.

* Empirical results demonstrate both the effectiveness and stability of the proposed method. Moreover, it is computationally efficient, introducing only a minor additional runtime overhead. The approach is also compatible with existing LoRA setups without requiring any architectural modifications.

**Weaknesses:**

* The experimental settings and details are somewhat limited and unclear. First, how are the experiments on the QA datasets conducted? Is the model fine-tuned on a mixed training dataset and then evaluated on several benchmarks, or is it fine-tuned on one QA dataset and tested accordingly? If it is the latter case, I would suggest conducting additional experiments on general language understanding and dialogue datasets such as WizardLM to better assess the model’s generalization ability. Moreover, the experimental settings for the reasoning tasks are also limited.

* How many steps does Stable-LoRA require to reach its stable mode? I think the shrinkage of the LoRA matrix $A$ shares a similar intuition with the weight decay mechanism in AdamW. How about comparing Stable-LoRA with AdamW using a relatively large weight decay parameter when the model shows instability?

* Finally, Stable-LoRA should also be applicable to other LoRA variants, such as AdaLoRA. Have the authors tried this? If the method generalizes well to other setups, it would be quite interesting.

**Questions:**

See details in the weaknesses part.

---

> ### Author Response · Authors · 2025-11-21
>
> We sincerely thank you for the review!
>
> > The experimental settings and details are somewhat limited and unclear. First, how are the experiments on the QA datasets conducted? Is the model fine-tuned on a mixed training dataset and then evaluated on several benchmarks, or is it fine-tuned on one QA dataset and tested accordingly? If it is the latter case, I would suggest conducting additional experiments on general language understanding and dialogue datasets such as WizardLM to better assess the model's generalization ability. Moreover, the experimental settings for the reasoning tasks are also limited.
>
> The model is fine-tuned on each dataset and tested accordingly, as we believe that LoRA is often used to solve a task-specific problem, rather than for a generalized purpose (which is more related to pretraining or SFT).
>
> Following your suggestion, we conducted experiments on a combined dataset of the 5 datasets in the original experiment. We choose 8k samples from each dataset and train 2 epochs on this combined dataset. The averaged accuracies of 5 datasets are shown in the table below. As demonstrated, Stable-LoRA can also show superiorities in this regime.
>
> | Model | 0.5B | 1B | 1.5B | 3B |
> | --- | --- | --- | --- | --- |
> | AdamW | 61.54 | 71.90 | 80.17 | 83.68 |
> | Stable-LoRA | 62.46 | 72.80 | 80.75 | 84.13 |
>
>
> > How many steps does Stable-LoRA require to reach its stable mode?
>
> Stable-LoRA reaches stability when the stable condition is met, while the actual number of step depends on the model type, dataset, hyperparameters and shrinkage ratio $\lambda$. You can see in Fig2 that, the step numbers are different under various $\lambda$ s, e.g. 200-400 steps for v_proj. Essentially the number is smaller with larger $\lambda$ since the shrinkage is faster.
>
> > I think the shrinkage of the LoRA matrix $A$ shares a similar intuition with the weight decay mechanism in AdamW. How about comparing Stable-LoRA with AdamW using a relatively large weight decay parameter when the model shows instability?
>
> The shrinkage operation is numerically equivalent to applying an extremely large weight decay rate to $A$ in the early stages (e.g. $w=50.0$ is numerically equivalent to $\lambda=0.005$ when $\eta=1e-4$). However, as we have explained in Appendix A, the motivation, scope and application schedule of the two methods are fundamentally different. The shrinkage is to prevent **instable feature learning**, and is only applied to early steps and merely on $A$. Instead, weight decay is to prevent **overfitting**, so it is applied during the whole of training on both $A$ and $B$. The shrinkage is much faster than weight decay, with ratios of $\lambda\approx 1e-3$ vs. $\eta\times w \approx 1e-6$, so the numerical effect of weight shrinkage will be much more significant than weight decay.
>
> Notably, Stable-LoRA is orthogonal to weight decay, and we have already applied weight decay $w=0.01$ across all experiments, including Stable-LoRA and AdamW.
>
>
> > Finally, Stable-LoRA should also be applicable to other LoRA variants, such as AdaLoRA. Have the authors tried this? If the method generalizes well to other setups, it would be quite interesting.
>
> Stable-LoRA is targeting at LoRA itself, while current popular LoRA variants tend to substantially change the formulation of $W=W_0+BA$. For example, AdaLoRA changed it to $W_0 + P\Lambda Q$, and DoRA changed it to $m\frac{W_0+BA}{||W_0+BA||_c}$. These modifications to LoRA also lead to essential changes of the training dynamics, as the learned feature $\Delta(Y)$ is no longer the same as in Equation (1). Therefore, it is infeasible to combining Stable-LoRA with these substantially different variants.

---

### Official Review · Reviewer_1UrF · 2025-11-01

**Soundness:** 2
**Presentation:** 2
**Contribution:** 2
**Rating:** 2
**Confidence:** 3

**Summary:**

The paper investigates whether LoRA can achieve and sustain stable feature learning. By introducing a $\gamma$-function as the main analytical tool, the authors prove that stable feature learning is attainable under specific hyperparameter settings and when $A = B = 0$. However, in practices, $A$ and $B$ cannot be set to 0 at the same time. To mitigate this issue, the paper proposes a progressive shrinkage strategy that gradually reduces $A$ during the early training stages. Extensive experiments suggest that the proposed Stable-LoRA method consistently outperforms baseline models.

**Strengths:**

* The paper introduces the $\gamma$-function as a novel analytical construct for understanding LoRA’s stability behavior.
* Theoretical analysis yields an interpretable condition—$A=B=0$—under which stable feature learning can be achieved.
* The proposed progressive-shrink mechanism is a practical solution to approximate the ideal condition $A=B=0$ and address the constratins that in practices, we cannot set both $A$ and $B$ to 0.

**Weaknesses:**

* Definition 1 lacks rigorous justification. While it aligns with prior empirical observations, it represents only one possible stability condition. The framework built upon it may have limited generalizability, which can only be supported by broader empirical studies.
* The definition of the $\gamma$-function appears mathematically infeasible. Although it seems inspired by logarithmic properties (e.g.,
$\log(x) + \log(y) = \log(x \times y)$ and $\log(x + y)$ is dominated by $\max(\log(x), \log(y)$). The equation $\gamma[v + v'] = \max(\gamma[v], \gamma[v'])$ may not always hold, even with a hidden constant in the $\Theta(\cdot)$. Thus, the $\gamma$-function may be valid for qualitative reasoning, but not for formal proof.
* Empirical results (e.g., Table 1) show only moderate improvements without reporting standard deviations or confidence intervals. It is therefore difficult to determine whether the observed gains are statistically significant or within expected variance.
* The writing quality could be improved. Several sections are hard to parse and would benefit from clearer exposition and more precise mathematical notation.

**Questions:**

1. Line 47: Should the asymptotic complexity be $O(n)$ instead of $O(n^0)$,  if this is "with respect to model width $n$" as stated in line 46.
2. Line 131: The definition of $r[v]$ by $v=\Theta(n^{\gamma[v]})$ is kind of informal. Written in this way, it implicitly invites a proof of existence.
3. Line 132: $\gamma[\overrightarrow{v}]:=\max(v_i, 0 < i < k)$. Should it be $0\le i$?

---

> ### Author Response · Authors · 2025-11-21
>
> We sincerely thank you for the review!
>
> > Definition 1 lacks rigorous justification. While it aligns with prior empirical observations, it represents only one possible stability condition. The framework built upon it may have limited generalizability, which can only be supported by broader empirical studies.
>
> Definition 1 is widely used in related works [1-3]. It serves as a foundation of analyzing stable feature learning conditions with respect to **model width**, rather than all possible contributing factors.
>
> We agree that training dynamics are influenced by many factors. However, our work focuses precisely on the factor of model width. Therefore, the purpose of Definition 1 is not to establish a universal condition including all factors (which is never done by any works and seemingly impossible), but to demonstrate the specific influence of model width and identify the instability issues associated with it. Existing works upon stable feature learning of LoRA [1-3] also mainly focus on the model-width factor, showing that it is indeed quite influential to fine-tuning performances. We never claimed that we have achieved a universal stable condition considering all factors simultaneously, but only about optimizing the single factor (model width) which already yielded meaningful gains. Other factors are certainly important as well, as discussed in related works (Sec. 6), and we encourage future research to explore them in depth.
>
> > The definition of the $\gamma$-function appears mathematically infeasible. Although it seems inspired by logarithmic properties (e.g., $log(x)+log(y)=log(x\times y)$ and $log(x+y)$ is dominated by $max(log(x),log(y))$). The equation $\gamma(v+v')=max(\gamma(v),\gamma(v'))$ may not always hold, even with a hidden constant in the $\Theta$. Thus, the $\gamma$-function may be valid for qualitative reasoning, but not for formal proof.
>
> The definition of $\gamma$-function and the equation $\gamma(v+v')=max(\gamma(v),\gamma(v'))$ are widely used in related works [1-3]. They are not our novel creation. In fact, the $\gamma$-function is introduced merely as a concise expression that $v=\Theta(n^{\gamma(v)})$, having no self-introduced new concept.
>
> The value of $\gamma$-function essentially reflects the most dominant term in the asymptotic scale of a given expression, the same as $\Theta$-notation. For example, consider $M=c_0n^3+c_1n^2+c_2$ where $c_0,c_1 \neq 0$ and $c_2$ are constants. We clearly have $\Theta(M)=\Theta(n^3+n^2)=\Theta(n^3)$, since the $n^2$ term is dominated and by $n^3$ as $n$ scales, and therefore $n^2$ is eliminated from the $\Theta$ expression. Accordingly, using $\gamma$-function for notation gives $\gamma(M)=\gamma(n^3+n^2)=\gamma(n^3)=3$. Consequently, the use of the maximum operator naturally follows from the properties of the $\Theta$-notation, which captures the highest-order term in asymptotic behavior.
>
> We hope that these clarification and example can clearly demonstrate that the definition of $\gamma$-function is mathematically rigorous, and explain why $\gamma(v+v')=max(\gamma(v),\gamma(v'))$ always hold, with constant terms having no effect on it.
>
>
> > Empirical results (e.g., Table 1) show only moderate improvements without reporting standard deviations or confidence intervals. It is therefore difficult to determine whether the observed gains are statistically significant or within expected variance.
>
> All the results we have reported are the averaged numbers under 3 runs, as stated in sec5. It is an effective and commonly-known form of statistical significance for showing empirical results in this area.
>
>
> > Line 47: Should the asymptotic complexity be $O(n)$ instead of $O(n^0)$, if this is "with respect to model width $n$" as stated in line 46.
>
> It is indeed $\Theta(n^0)$. We show that LoRA can naturally achieves stable feature learning with respect to model width, and stable feature learning is defined in Definition 1 as $\Theta(\Delta(Y))=\Theta(n^0)=\Theta(1)$.
>
> > Line 131: The definition of $\gamma[v]$ by $v=\Theta(n^{\gamma[v]})$ is kind of informal. Written in this way, it implicitly invites a proof of existence.
>
> As stated above, $\gamma$-function is just a concise notation. We believe that all the expression $Y$ can naturally be quantified by $\Theta$ to show its asymptotic behavior w.r.t. n, and therefore the use of $\gamma$-function is sound.
>
> > Line 132: $\gamma[\overrightarrow{v}]:=\max(v_i, 0 < i < k)$. Should it be $0\le i$?
>
> Yes, it is a typo, and we will fix it in the new version.
>
> [1]Lora+:Efficient low rank adaptation of large models.In International Conferenceon Machine Learning, pp. 17783–17806. PMLR, 2024.
>
> [2]Riemannian preconditioned lora for fine-tuning foundation models.In Proceedings of the 41st International Conference on Machine Learning, pp. 59641–59669, 2024
>
> [3]The impact of initialization on lora finetuning dynamics. Advances in Neural Information Processing Systems, 37:117015–117040, 2024.

---

> > ### Comment · Reviewer_1UrF · 2025-11-25
> >
> > Thank you for the detailed response. The discussion on the $\gamma$-function is sound and helped dispel my major concerns. I will increase my score to 4.

---

> > > ### Author Response · Authors · 2025-11-25
> > >
> > > Thank you for reading our response, and it is good to know that the concerns have been addressed.
> > >
> > > If you have any further questions to discuss, please feel free to let us know. Thank you!

---

### Official Review · Reviewer_33Pn · 2025-11-01

**Soundness:** 3
**Presentation:** 2
**Contribution:** 3
**Rating:** 4
**Confidence:** 3

**Summary:**

LoRA is the method that is the most commonly use to finetune LLMs. However it does have some training challenges. This study looks at those problems and proposes a clear analysis, especially around the initialization of LoRA--and propose a novel solution. Empirical results are provided to support the claim.

** Strength **
- very important problem in the world of LLM and post-training especially
- the experiments are quite sound to validate the study
- the interpretation of LoRA is interesting albeit hard to grasp for readers not familiar with the field

** Weakness **
- enormous typo (e.g. one right in the title, adaption, or in the abstract, Langauge) which makes the entire script feel quite odd
- could the author discuss a possible link with https://arxiv.org/pdf/2410.09692? who also checked at initialization and training dynamics?
- lacking some geometric/visual intuition that would help reader better grasp the results
- needs better scoping of limitation and future work

**Strengths:**

Please see summary

**Weaknesses:**

Please see summary

**Questions:**

Please see summary

---

> ### Author Response · Authors · 2025-11-21
>
> We sincerely thank you for the review!
>
> > could the author discuss a possible link with https://arxiv.org/pdf/2410.09692? who also checked at initialization and training dynamics?
>
> The paper you mentioned is Allora: Adaptive Learning Rate Mitigates LoRA Fatal Flaws. Allora also discussed about the impact of zero initialization of LoRA on fine-tuning performances (along with other LoRA components like dropout and scaling factor).
>
> The main difference is that, the two papers considers the impact of zero init in different aspects:
>
> Allora considered it as a short-term and gradient-oriented issue. It focuses on the phenomenon that, the zero initialization of $B$ would result in the gradient of $A$ ($g_A$) to be zero at beginning, while $g_B$ not zero. This leads to imbalanced training between $A$ and $B$ ($B$ is effectively trained while $A$ is not). Although the imbanlance only concerns the first few steps, it will be a severe problem in finetuning regimes where the total number of training steps is small.
>
> Our paper considered it as a long-term and output-oriented issue. We identify that the problem of instable feature learning will exist in the whole training process, as mentioned in sec4. And our analysis are all based on the output feature learning $\Delta Y_t$ (equation (1) in our paper), the expression of which includes $g_A$ and $g_B$ for sure. We will cite Allora for the above clarifications.
>
>
>
> > lacking some geometric/visual intuition that would help reader better grasp the results
>
> The motivation of this paper largely lies in theoretical analysis of stable feature learning, which is highly related to training dynamic. We believe that these theoretical analysis and results can be better demonstrated by formulas and mathematical induction, while a figure demonstation may lack rigorousness and even confuse the readers.
>
> We do provide visual analysis of averaged norm value of $A$ and $B$ in Fig.2, which is an empirical justification for the existence of the instability of feature learning, emphasizing our motivation.
>
> > needs better scoping of limitation and future work
>
> The limitations are in Appendix E, where we discussed the limited scope of experiments about LoRA variants, initialization strategies and optimizer. The future work will target at these limitations.
>
> Lastly, we appreciate that you pointed out the typos, and we will fix them in the new version.

---

### Author Response · Authors · 2025-12-03
**Final remark from authors**

Dear Reviewers, AC and SAC,

Thank you for handling our submission!

We greatly appreciate the reviewers' thoughtful comments and suggestions, and we have adequately solved the raised concerns in our rebuttal. We sincerely thank you for reading our rebuttal and providing feedback.

Before the discussion phase concludes, we would like to note a few points for consideration in the decision-making process:

### 1. Summary of contributions

Our paper (1) theoretically shows that LoRA can naturally achieve stable feature learning, while also showing that the empirically necessary non-zero initializtion of $A$ may compromize this stability; (2) proposes a weight-shrinkage optimization for $A$ during the early stage of training, to mitigate the potential instability.

### 2. About the theoretical concerns

(1) Definition 1 (about stable feature learning) has been widely adopted and explored in current works [1-3]. It focuses on the impact of model width on feature learning, a crucial aspect as model parameter counts continue to grow.

(2) The induction and theoretical derivations are mathematically sound and rigorous. The concerns raised by Reviewer 1UrF and iYHY have been fully clarified.

(3) We believe that a formulated mathematical induction provides a clearer and more precise presentation of the training dynamics, rather than visual demonstrations. We have also explicitly summarized the motivation and method in Fig.1 and Fig.2.

### 3. About the empirical concerns

(1) We followed the suggestion of Reviewer iYHY and conducted **additional comparison with more baselines**, and the results consistently support our conclusions.

(2) We clarified (in our response to Reviewer S7AS) why Stable-LoRA is **not applicable** to substantial variants of LoRA, due to their fundamentally altered training dynamics.

(3) We followed the suggestion of Reviewer S7AS and conducted extensive experiments on a **task-combined dataset**, which demonstrate the **generalization capability** of Stable-LoRA.

Thank you again for your time and for considering these remarks during the final discussion.

Best regards,

Authors of Paper #6237

---

[1]Lora+:Efficient low rank adaptation of large models.In International Conferenceon Machine Learning, pp. 17783–17806. PMLR, 2024.

[2]Riemannian preconditioned lora for fine-tuning foundation models.In Proceedings of the 41st International Conference on Machine Learning, pp. 59641–59669, 2024

[3]The impact of initialization on lora finetuning dynamics. Advances in Neural Information Processing Systems, 37:117015–117040, 2024.

---

### Meta-Review · Area_Chair_nUmd · 2026-01-07

**Summary:**

The paper proposes Stable-LoRA, a method to mitigate feature learning instability in LoRA caused by non-zero initialization, utilizing a progressive weight-shrinkage strategy. Reviewers generally appreciated the theoretical motivation identifying the stability issue and the simplicity of the proposed solution. Initial concerns focused on the rigor of the theoretical definitions (specifically the $\gamma$-function), the scope of baselines, and the scale of experiments. The authors provided a robust rebuttal, including new experiments on larger models and additional baselines.

**Reviewer Concerns:**

Addressed by Rebuttal:
- Theoretical Rigor: The authors clarified the definition and mathematical validity of the $\gamma$-function and Definition 1.

- Experimental Scope: The authors conducted additional experiments on larger models (Llama-3.1-8B), combined datasets for generalization, and comparisons against stronger baselines like DoRA.

- Methodological Clarifications: The authors clarified the distinction between their method and standard weight decay and explained why the method is not directly applicable to structural variants like AdaLoRA.

Outstanding:

- Significance: While the method is effective, reviewers see the improvements as a bit incremental rather than transformative, despite the successful rebuttal.

**Reviewer Scores:**

one reviewer raised from 2 to 4. I would expect another 4 to 5. And one of the review (score of 4) is quite low quality and I don't consider it in the final decision. I think the final score as 456 which is borderline

---

### Decision · Program_Chairs · 2026-01-26

Accept (Poster)